# Decoding China’s COVID-19 Health Code Apps: The Legal Challenges

**DOI:** 10.3390/healthcare10081479

**Published:** 2022-08-05

**Authors:** Xiaohan Zhang

**Affiliations:** School of Law, Zhejiang University City College, Hangzhou 310015, China; zhangxiaohan@zucc.edu.cn

**Keywords:** COVID-19, health code apps, personal information, data protection, contact tracing, ethics, privacy, personal information protection law

## Abstract

Heath code apps, along with robust testing, isolation, and the care of cases, are a vital strategy for containing the spread of the COVID-19 outbreak in China. They have remained stable and consistent, allowing China to extensively restore its social and economic development. However, the ethical and legal boundaries of deploying health code apps for disease surveillance and control purposes are unclear, and a rapidly evolving debate has emerged around the promises and risks of their fast promotion. The article outlines the legal challenges by applying the core values of the Personal Information Protection Law (PIPL), the fundamental law for personal information protection in China, into the context of the nationwide use of health code apps. It elaborates on the balance between the demands for upholding individuals’ rights to the security of their personal information and those for public access to such information to prevent the spread of infectious diseases. It identifies the current gaps in addressing personal information harms during the use of the apps, particularly with regard to user consent, transparency, necessity, storage duration, and security safeguards.

## 1. Introduction

The outbreak of the COVID-19 pandemic has triggered an unprecedented public health crisis around the world, with varying degrees of impact on the political, economic, cultural, and other social spheres of countries. Globally, as of July 2022, more than 571 million confirmed cases of COVID-19, including more than six million deaths, were reported to the WHO [1]. In response, digital technologies were harnessed or proposed to fit the newly emerged expectations, with contact tracing apps being a common tool [2,3]. The contact tracing apps are not able to bring the pandemic under control by themselves. However, the evidence demonstrates that they are useful, providing that they have adequate political backing and are appropriately integrated into the public-health systems [4].

In China, the deployment of health code apps, which aimed to help governments identify the people who were potentially exposed to COVID-19, has promised to bring forth increased accountability, quality, efficiency, and innovation [5]. The citizens had to follow similar steps to register and obtain their health codes through software extensions on the Alipay app or the WeChat app, regardless of the province they live in. While not technically compulsory, a clean result (i.e., a green code) must be presented to access certain public areas, such as schools, restaurants, malls, hospitals, and public transportation stations [6]. Therefore, the national health code system has become an innovation that no ordinary citizen can live without in pandemic-era China [7].

Since the COVID-19 pandemic, contact tracing apps have garnered much attention from academics and governments. Extensive debates over issues concerning privacy have primarily focused on Western technologies and experiences, particularly those developed by Google and Apple [8,9,10,11,12]. However, the health code apps in China differ greatly from other contact tracing apps in significant ways. Firstly, rather than standalone apps, the health code apps are software extensions of the existing WeChat and Alipay apps, which have already penetrated the lives of the Chinese people. Since both WeChat and Alipay already possess a wealth of personal data, being embedded in these apps makes the health code apps more direct and intrusive. Furthermore, the Chinese tech businesses have less bargaining power with the government over the control of data flow compared to their Western counterparts [13].

The ubiquitous health codes have posed considerable legal challenges under the current Chinese regulatory framework. The answers to many burning questions remain ambiguous, even over two years after the health code apps were nationally rolled out. What kind of personal information is being collected? Who is authorized to access the data? How are the data processed and stored? What happens to the data collected after the outbreak is over? What safeguards have to be in place to mitigate the risks at stake? An increasing number of people are raising concerns about the ethical and legal risks of the health code system. It is thus crucial to ensure that the use of such innovation reflects, at all times, a fair balance between the various public interests and individuals’ fundamental rights protected by the law, and is proportionate to the legitimate purpose pursued.

This article begins by presenting the astonishing promotion speed of the health code apps as a powerful method for curbing the spread of the virus and then turns to the concerns they raised in these contexts. It focuses on a prominent (but not the only) worry: the threats to personal information. By looking into the Hangzhou Health Code System case, it elaborates on what information is required for the generation of individual health codes, what databases are accessed, and how a health code system basically works. The following section highlights the main principles and provisions under the Personal Information Protection Law (PIPL) as China’s basic law for information protection. Section 4 further examines the legal and technical requirements consistent with those principles and the gaps in compliance. The overall aim of this paper is to provide insights into how to better achieve the equitable and appropriate use of health code apps.

## 2. Materials and Methods

### 2.1. Research Data

While extensive fieldwork is required to adequately investigate health code apps, this remains challenging, given the current pandemic situation. Accordingly, regarding the facts about the deployment and development of health code apps, the relevant information and data were collected mainly through the official websites of the involved institutions, such as the National Health Commission of China, the State Council of China, the National Bureau of Statistics of China, the China Internet Network Information Center, the Government of Zhejiang Province, and other public sources, such as the domestic surveys conducted by influential publishers (for example, the survey conducted by the People’s Daily on the usage and public acceptance of health code apps), as well as reports and commentaries in Chinese and non-Chinese news media. The search period for the factual data is set between February 2020 and June 2022.

For the regulations, policies, and specifications about the health code management, the research materials and the data were collected mainly through the relevant governmental departments, such as the State Administration for Market Regulation of China, Zhejiang Provincial Administration for Market Regulation, and Hangzhou’s Center for Disease Control and Prevention. For the academic analysis and discussions of the technical and ethical issues regarding the COVID-19 contact tracing apps, the research literature and references were mainly obtained by searching databases, such as Web of Science, PubMed, and CNKI using keywords COVID-19, contact tracing, and health code.

For the research questions on the governing laws and precedent cases, such as the interpretation and application of the relevant provisions of the EU General Data Protection Regulation (GDPR), the PIPL, and the OECD Principles on the Protection of Privacy, the references were acquired by searching HeinOnline, Westlaw, PKUlaw, LexisNexis, and other professional legal databases.

### 2.2. Research Methods

Diverse versions of health code systems were implemented in China’s provinces and municipalities. This article mainly delves into the case of Hangzhou, one of the first cities to adopt the health code apps for COVID-19 prevention and control. This representative case is worth examining in depth to figure out how a health code system is designed and intended to work, what databases would be accessed, what personal information will be collected, and how the codes of different colors are assigned. The ethical and regulatory issues, whether and to what extent health code apps are reliable, whether the data collection measures comply with the current legal regime, and whether the rights of people are sufficiently protected are all based on the central questions addressed in the case studies.

Interdisciplinary methodologies were adopted, including empirical research of the law, technology, policy science, and health studies. A comprehensive search and literature review of the development of the health code apps were conducted to provide a solid basis for the empirical analysis and further discussions.

Doctrinal research is fundamental to the study of legal issues. The article identifies, describes, and critically analyzes the principles and provisions contained in the PIPL, the Civil Code, and the other related laws and regulations of China, as well as relevant regulations of different regions and countries. The textual analysis aims to facilitate the understanding of the legal risks in the widespread use of health code apps, based on which improvement suggestions and solutions are provided.

## 3. Results

### 3.1. The Course That Health Code Apps Went National

At the commencement of the COVID-19 outbreak, measures of mass immobilization were implemented across the country, including the restriction of interregional movement, the closure of hard-hit areas, and the close-off management of residential clusters [5]. Such severe and far-reaching policies quickly appeared untenable and unsustainable. The need for a delicate balance between economic recovery and epidemic control demanded the development of more precise and fine-tuned technologies for tracking the virus’s progress more effectively and monitoring the health state of the population on a grander scale. The health code apps were rolled out as a result of this predicament. They quickly appeared to have handled various pandemic control challenges in China.

The health code apps initially debuted in Hangzhou, the home base of Alibaba Group, in early February 2020. Hangzhou established a task force consisting of the Economic and Information Technology Commission, the Health Commission, the Big Data Bureau, the Development and Reform Commission, and other departments on 6 February 2020, to activate the health code program for the extraordinary task of the epidemic prevention and control [14]. On 11 February, Hangzhou Health Code, a mini app integrated with Alipay, was officially launched, which indicates that its entire creation process only took four days.

Alibaba supplied the technology for the health code apps, including the display page design and algorithm, among other things. The Hangzhou Municipal Government oversaw the database’s management. Within a few weeks of the Hangzhou trial, it was claimed that over 200 cities in China had partnered with Alibaba, Tencent, and other big ICT service providers to implement similar initiatives in their jurisdictions [8]. The health code apps were promoted as quickly as around one month from the implementation in Hangzhou City, Zhejiang Province, to the entire country. More precisely, all of the 31 Chinese provincial governments adopted health code apps in only 39 days [15]. Figure 1 clearly demonstrates how rapidly health code apps were rolled out in China.

### 3.2. Data Collected for Generating a Personal Health Code

Based on the national standard documents [16], the information acquired from the official websites, news sources, and interviews, the data that might be needed to generate a personal health code are described in Table 1. All the data could be broadly classified into four categories, as shown in Figure 2. Although some of the data may not be retrieved in the actual process, basic personal information, health status, and travel records, among other things, are required at the very least to be in place.

### 3.3. Databases That a Health Code System Can Access—The Hangzhou Health Code System as an Example

The local governments are in charge of the health code apps in various areas of China, with the majority of them partnering with Alibaba, Tencent, or other technological businesses, and their systems operating similarly. A health code system must access numerous databases to achieve its operation. Taking the Hangzhou health code system as an example, the Health Code Management Specifications for Infectious Disease Prevention and Control Personnel (hereafter referred to as “Specifications”) issued by the Zhejiang Provincial Market Supervision Administration listed all of the databases that the system can access. According to the Specifications, the generation of a Hangzhou personal health code is based on the data stored on the Provincial Public Data Platform, which is administrated by the Provincial Big Data Bureau.

The Provincial Public Data Platform has access to several databases that store citizens’ data, including the province-wide database of “key populations”, the national close-contact database, the regional risk classification database, the personnel location database (which collects location data retrieved from China’s three major telecommunications operators—China Unicom, China Mobile, and China Telecom), and the province-wide health code-sharing database. The Specifications did not further specify who precisely were defined as “key populations”, who was in charge of the databases, or what information each database indeed covered.

Furthermore, the Hangzhou Health Code system was also authorized to access other essential databases for epidemic prevention and control, such as the comprehensive database of population, the comprehensive database of legal persons, the database of electronic certificates, the database of credit information, and the database of geographical information, according to Article 7.2.1.1 of the Specifications. It is also worth noting that the databases of foreign nationals were required to be established in each of the administrative regions to record information including foreigners’ names, genders, home countries, passport numbers, places of residence, contact numbers, whether they have visited infected areas, whether they have contacted with people from infected areas, and so on.

Based on real-time automated health assessments, coupled with the function of roaming and positioning across the provinces and cities, the system would generate a QR code in one of three colors: green for unrestricted travel; yellow for mandatory quarantine for seven days at designated places (home, hotel, or other accommodation); and red for confirmed or suspected cases that must be treated or isolated/quarantined for fourteen days [17]. Figure 3 illustrates how the Hangzhou health code system creates an individual health code.

### 3.4. Public Acceptance of the Health Code Apps

Combined with containment measures, such as top-down restrictions on movement and traditional epidemiological legwork, the use of the health code apps shows both China’s ability to collect vast amounts of data as well as the ability to make relatively free use of it [18]. Promoting such a system in the West may appear to be a near-impossible mission. For example, Twitter just agreed to pay a $150 million fine after federal law enforcement officials accused it of illegally using peoples’ data, highlighting the sensitivity of personal data collection [19]. In contrast, the general public in China appears more prepared to surrender personal information in return for convenience and a normal order of work and life. According to a People’s Daily poll conducted in July 2020, with 5928 valid questionnaires collected, 90% of people had applied for their health codes at the time and were generally satisfied with its role in the epidemic prevention and control. People awarded an average score of 8.49 out of 10 when evaluating the function of the health code apps [20]. Figure 4 indicates people’s high acceptance of the health code apps.

As the preliminary analysis of the UK National Health Service (NHS) Test and Trace program showed, contact tracing app interventions can have a considerable impact on the epidemic control. The group established the 56% adoption rate metric after researching the effectiveness of contact tracing apps [21]. This metric then became the most commonly cited adoption rate in the literature, with the WHO later noting that a 60–70% adoption rate was needed [22]. According to the 49th Statistical Report on Internet Development in China, released by the China Internet Network Information Center (CNNIC), as of December 2021, the cumulative number of users of the health code apps nationwide exceeded 900 million, and the cumulative number of visits exceeded 60 billion [23]. Within 72 h of the Hangzhou Health Code’s implementation, the application rate for the Hangzhou Health Code even reached 95.2% in some of the localities [24]. The high adoption rate in China, combined with the relatively wide public acceptance, has been key to enabling massive quantities of data to be collected and put to use across China, which in turn vastly enhanced the effectiveness of the Chinese health code apps [25].

## 4. Discussion

Pandemics put society under immediate but temporary strain to adapt and develop new health solutions. The actions that would not be suitable in other circumstances are frequently permissible in the context of a public health emergency [26,27]. The health code apps, for instance, were put out nationwide in less than a month. In this sense, immature technologies and designs were likely to have been pressed into service in response to the public emergency. Due to the potential risks, several objections were raised over time, with the most pressing one being the issue of the personal information of individuals. Although it appears that certain invasions of personal information may be justifiable if they have the potential to save many lives and alleviate considerable suffering, oversight mechanisms are still needed to set restrictions on the usage of such fast-evolving applications.

### 4.1. The Introduction of the PIPL

The health code apps revealed severe challenges to data governance and flaws in the existing regulatory framework. These problems, in turn, encouraged public awareness about privacy and personal data protection and motivated law-making processes to catch up and regulate the use of digital technologies. Nearly two years after the COVID-19 outbreak, China’s first comprehensive data protection law, the PIPL, came into force on November 1, 2021. Combined with two other paramount pieces of cybersecurity and data security legislation, the Cybersecurity Law (CSL) and the Data Security Law (DSL), it is remarked to have established a new data protection regime for China [28]. The implementation of the PIPL could provide new insights into the legal and appropriate use of the health code apps.

### 4.2. Seven Key Principles Set out by the PIPL

The PIPL is modeled in part on the EU’s GDPR. As a fundamental law for personal information protection in China, the PIPL clarifies the rules for processing personal information, the obligations of personal information handlers and processors, and the rights of personal information subjects. It sets out seven key principles for safeguarding personal information, including lawfulness (Article 5), purpose specification (Article 6), data minimization (Article 6; Article 19), transparency (Article 7), accuracy (Article 8), accountability (Article 9), and data security (Article 9; Article 51). The implications corresponding to each of the principles are demonstrated in Table 2.

These seven principles outlined by the PIPL raise legal considerations that are both cross-sectional and domain-specific. Related legal challenges can be identified by translating the principles and values into the context of health code apps.

### 4.3. Gaps in Addressing Personal Information Harms

#### 4.3.1. User Consent

The consent of the data subjects prior to processing their personal data is an essential precaution in data protection. The data controller shall fulfill the obligation to inform and obtain individuals’ explicit and specific consent before processing their personal information [29,30]. This requirement is widely applied in personal information protection legislations and recommendations around the globe. For example, under the GDPR, processing personal data is generally prohibited unless it is expressly permitted by law or the data subject has consented to the processing. The foundational 1980 OECD Principles on the Protection of Privacy, including the principles of Collection and Use Limitation, and Security Safeguards and Individual Participation, require that any use of personal data be made with the knowledge and consent of the data subject, where possible [31]. In China, Article 12 of the PIPL, Article 1035 of the Civil Code, and Article 41 of the CSL all clearly set user consent as a premise for the processing of personal information.

Nevertheless, user consent is not required under all circumstances. It is also a common practice in legislations to exempt public interest or public health from the premise of the data subject’s consent. For example, Article 9 of the GDPR provides that public agencies could lawfully collect and analyze data concerning health, pursuant to Articles 9.2(g) (‘substantial public interest’), 9.2(i) (‘preventative medicine’), and 9.2(h) (‘public interest in the area of public health’).

In the PIPL, Article 13 states that an individual’s consent is not required if the processing is necessary to respond to public health emergencies or to protect people’s life, health, or property safety in an emergency. In addition, article 1035(1) of the Civil Code stipulates that the personal information of a natural person can be processed without the person’s consent “as prescribed by laws or administrative regulations”. In the event of a public health emergency, China’s Emergency Response Law, Law on Prevention and Treatment of Infectious Diseases, as well as Regulation on Responses to Public Health Emergencies all authorize the public agencies to collect personal information without the data subject’s consent. To put it another way, user consent is just one of the legal bases to justify the collection, handling, and storage of personal data in China. Public health causes can also be a proper legal basis for the authorities to obtain data without user consent amid a pandemic.

#### 4.3.2. Transparency

Real-time access to epidemic information is critical for making accurate decisions and adopting proper preventive measures in the face of epidemic prevention and control. Therefore, obtaining personal information without the data subject’s consent for public health might be justified during an emergency [32]. However, the exemption from the principle of user consent does not negate the necessity for procedural justice in processing personal data. Instead, more cautious and unambiguous legal norms for data protection must be respected, due to the significant potential for the misuse of governmental public power.

“No user consent required” does not exclude the data subjects from having the right to know. According to Article 7 of the PIPL, “personal information shall be handled under the principles of openness and transparency, with the regulations of personal information processing disclosed, and the aims, methods, and scope of processing openly declared.” Generally, this information should, in practice, be contained in a privacy notice. In many regions in China, however, the use of the health code apps indeed did not adhere to this principle. In October 2021, more than one and a half years after the nationwide rollout of health code apps, there were still fifteen health code apps used by various provinces and municipalities without any privacy notice. In several places where privacy notices were included, the notices suffered from problems with reading clarity and a lack of eye-catching prompts. Some of the provinces immediately copied the privacy notice of the “Suishen Code” app, the Shanghai version of the health code apps, with the date copied together incorrectly [33]. It appears that including privacy notices within the health code apps could be more of a formalistic requirement for many of the places. It is unclear whether it can indeed serve to improve transparency in the collection of personal data.

A lack of adequate transparency compliance measures can result in a number of issues, the most troublesome of which is the possibility of abuse of power. For instance, the legitimacy of the health code apps has been significantly undermined by recent acts in the central Chinese province of Henan. In June 2022, Henan Province and its capital, Zhengzhou City, were widely reported to be using the health code apps to prevent local bank clients from protesting against the potential loss of their savings in local rural banks on the brink of collapse [34,35]. According to the news reports, the health codes of over a thousand people were changed to red by city officials, so these people could not enter Zhengzhou City [36]. This shows that, without strict requirements on transparency, the health code apps can turn into a surveillance tool that enables government agencies to exert population control in the name of public health.

#### 4.3.3. Data Minimization

According to Article 6 of the PIPL, “personal data must be handled for a clear and reasonable purpose that is directly connected to the processing purpose, and in a manner that has the least impact on individuals’ rights and interests.” This provision draws on the “data minimization principle” provided by Article 5(1)(c) of the GDPR, which stipulates that personal data must be “adequate, relevant and limited to what is necessary in relation to the purposes for which they are processed.” It works in tandem with the GDPR’s “purpose limitation principle”, requiring that the purpose for which personal data is collected must be specified, explicit, and legitimate, as well as the “storage limitation principle”, stating that personal data should be kept no longer than is necessary for the purposes for which it is processed.

Under this principle, when personal information rights and interests must be restricted on the public health ground, it should be completed in the least harmful, least severe, and least frequent manner possible, with the scope of data subjects and the contents of the personal information collected being minimized as much as possible. Applying this principle to the context of the health code apps, the range of data collected should be restricted to the personal data relevant to and essential for achieving the purpose of the epidemic prevention and control. Such information may include basic information, such as name, address, ID number, and contact information; health information, such as body temperature and symptoms; as well as travel information, such as travel history and transportation records. However, it is not permitted to collect irrelevant or unnecessary personal data.

In this sense, the health code apps operated in many provinces and municipalities might have violated this principle. For example, the health code apps of Beijing, Shanghai, and Jiangxi Province mandate the collection of facial recognition data at the time of application. If the user does not consent, he or she will not be able to use the app. In cities such as Hangzhou, Wuhan, and Changsha, however, facial recognition information is not required for the use of the health code apps. This distinction, at the very least, demonstrates that the normal usage of the health code apps is unaffected by the absence of facial recognition information and that the collection of facial recognition data is not necessarily required to meet the goal of epidemic prevention and control.

In addition, facial recognition data are identified by Article 28 of the PIPL as “sensitive personal information”, of which the leakage or illegal use could easily lead to the violation of the personal dignity of a natural person, or harm to personal safety or property safety. Under Article 29, the authorities are only allowed to process sensitive personal information if they have specific purposes, sufficient necessity, and strict protection measures, with separate consent obtained before the data are processed. As such, the rules regarding the collection of sensitive personal information have been ignored in the areas where the compulsory collection of facial recognition data is required.

#### 4.3.4. Storage Duration

A significant concern for many people is not only their privacy today during the pandemic, but also their privacy in the future: whether full privacy protections will be restored once the pandemic is over, and whether the data collected today as a last resort will be used in unacceptable ways later. This underscores the need to include a compelling description of the time limit of any collection of personal data without consent. Ensuring that the authorities erase or anonymize personal data when they no longer need it will reduce the risk that it becomes excessive, inaccurate, or out of date.

The storage limitation principle is provided in Article 5(1)(e) of the GDPR, which states: “personal data shall be kept in a form which permits identification of data subjects for no longer than is necessary for the purposes for which the personal data are processed”. A comparable provision is included in Article 19 of the PIPL, which stipulates that “a retention period of personal information shall be the shortest time necessary to achieve the processing purpose, except as otherwise provided by any law or administrative regulation.” It indicates that when the state of emergency diminishes, collecting personal information becomes optional but not necessary for containing the epidemic. Any data that have already been collected should be discarded or made anonymous. However, most provinces and municipalities have not set specific time limits for data storage or any rules for erasing the personal information after the pandemic [33]. As such, the storage duration of personal information is still uncertain.

#### 4.3.5. Data Security

Individual citizens’ personal information may not be of great value or significance in and of itself. Still, the volume and scope of the personal data obtained during an epidemic make the information valuable and reusable. To guarantee the security of personal data, the least-damaging processing methods, such as “de-identification” or “anonymization”, shall be employed as much as feasible in data processing, as required by Article 51 of the PIPL [37].

However, the data collected by the health code apps are currently handled in a centralized manner, for instance, by being gathered and sent to the provincial Big Data Bureau, Alibaba, and the telecommunications department for processing [38]. Compared to the manual methods, a more significant number of entities, ranging from individual users to unauthorized parties/hackers, technology developers and owners, third-party services, public health agencies, and government entities, have the potential to access a tremendous amount of user information. Such user information includes but is not limited to user personal details, ID numbers, disease and exposure status, other health information, location information, social graph, and device information.

The centralized data management architecture enables data aggregation across multiple sources and subsequent user re-identification, especially when third-party services are engaged in the operation of the technology [39,40]. This was highlighted by a data leak that occurred in December 2020 in Beijing. Because of the flaws in the “Beijing Health Bao” system, the photographs, ID numbers, and nucleic acid test information of around seventy celebrities were made available for cheap online purchase [41]. This case demonstrates that there are still insufficient safeguards in place to maintain personal information security. In particular, when a third party other than the public authority may participate in data processing during an emergency, the use of data must be handled with greater caution [42].

## 5. Conclusions

Big data and computational revolutions have substantially promoted surveillance and social sorting. Since surveillance can easily cross the line between disease surveillance and population surveillance, the continuous monitoring and flexible adaptation to specific contexts must be in place. The speed at which the health code apps are being developed has outpaced the ability of the authorities to test their legality in due course. When the pandemic gradually subsides, the “special measures” once implemented during the state of emergency have to be thoroughly re-examined.

Sustainable innovation relies on taking a stakeholder approach to avoid systemic risk and optimize the outcomes [43]. Since the health code apps have great potential to reshape post-pandemic institutions, social orders, and daily life, the decision-makers have to ensure procedural robustness and limit ethical lapses while utilizing them. It is already time for the authorities to carefully consider the information protection implications within the apps, including but not limited to:
the legal basis for collecting and processing the personal data in these apps;whether the apps are implemented in a transparent, accountable, and comprehensive manner;whether the personal data gathering is necessary and proportionate, taking into consideration the amount of personal data collected, the specific purposes of the collection, and how the data will be processed and shared;the period within which these apps will be in effect, as well as whether the apps have made any reference to the data retention term;whether the data controller has taken strict management and technical measures to prevent a data leakage.

It is advised that the authorities adopt a more inclusive policy-making approach to address today’s and future’s most pressing challenges relating to personal data collection and processing. In doing so, the empowerment of people and the opportunities for innovation would rest on more solid foundations.

It should be noted that the pandemic situation makes it difficult to thoroughly research the health code apps using first-hand data, despite the need for substantial fieldwork. The findings here are primarily based on an analysis of the data from official websites, normative documents, and news media. Caution must be taken in the interpretation of the results.

The paper does not engage with satisfactorily resolving the challenges that have been identified. Given China’s social environment, this engagement should be completed in the context of specific technologies, healthcare systems, and political concerns. The PIPL does provide the administrative liabilities for violations of the law. In the meantime, civil liabilities, criminal liabilities, and other legal consequences are also mentioned thereunder. However, the relevant articles are only provided generally without details. The specific rules are still to be explored and implemented by local governments. This issue may be subject to further in-depth discussion for ongoing research.

Finally, unlike European countries, China does not have an independent data protection authority. Instead, several departments, such as the Cyberspace Administration of China, the Ministry of Public Security, and their local counterparts, have certain law enforcement powers over personal information protection problems. The PIPL did not alter the polycentric supervisory system. Future research may pay more attention to the need for, and methods of, establishing an independent data protection authority in China to comprehensively enforce the law.

## Figures and Tables

**Figure 1 healthcare-10-01479-f001:**
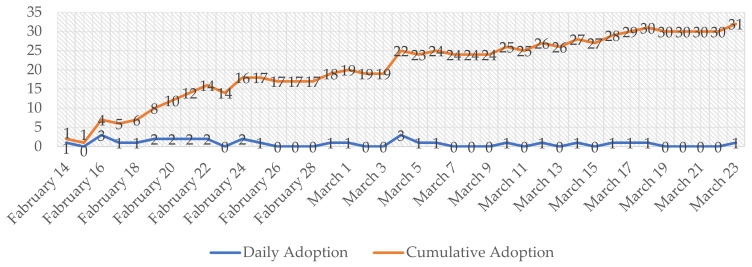
The number of provincial governments adopting health code apps.

**Figure 2 healthcare-10-01479-f002:**
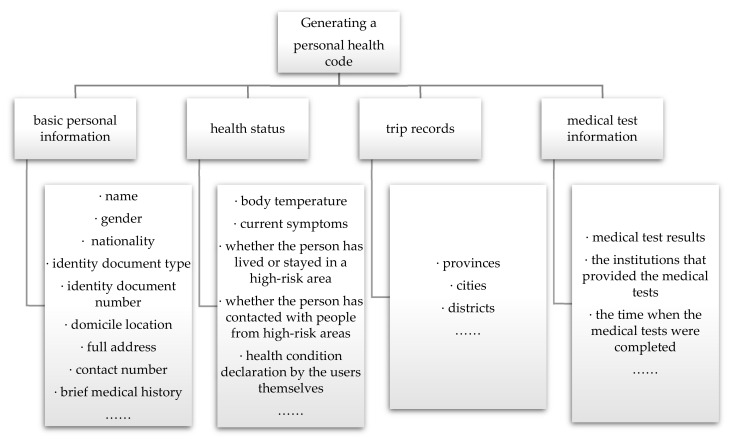
Categories of information required for generating a personal health code.

**Figure 3 healthcare-10-01479-f003:**
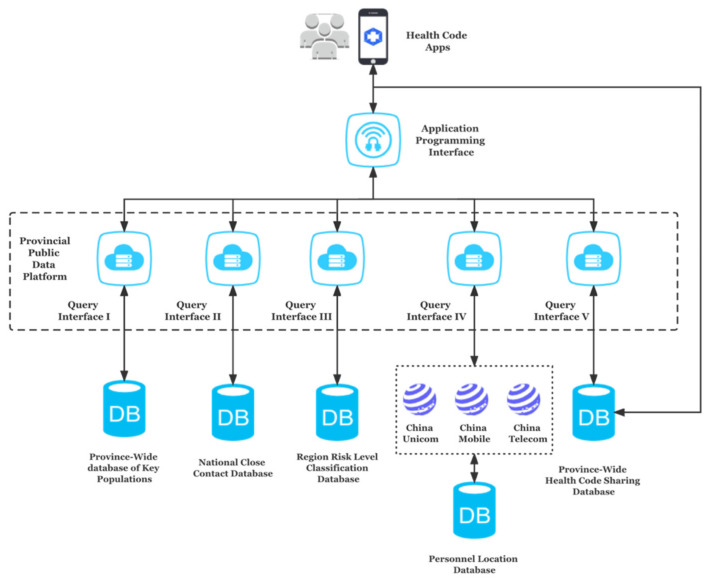
Hangzhou Health Code operation system.

**Figure 4 healthcare-10-01479-f004:**
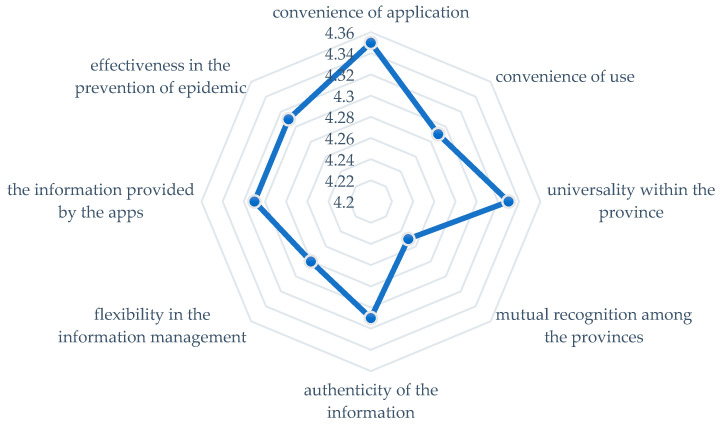
Public evaluation of health code apps (5-point scale).

**Table 1 healthcare-10-01479-t001:** Data that might be collected for generating a personal health code.

No.	Data Type	Illustrations
1	data of confirmed and suspected cases	name, age, and body temperature
2	data of close contacts	name, place of residence, and travel history
3	medical testing data	nucleic acid test results and antibody test results
4	data from fever clinic	name, body temperature, and time of fever onset
5	location tracking data	location information archived by the telecom operators
6	transportation information	train/flight/bus number, transfer time, and boarding and alighting time
7	information on immigration	arrival time, port of entry, and flight information
8	customs inspection data	item name, entry time, and test results
9	community data	place name, community name, and risk level
10	information recorded by the community	residence information, family member information, and residents’ travel information recorded by the community council
11	personal information collected in the public places	body temperature information and access records reported by each public place
12	information filled in by the users themselves	user-reported personal health data and brief medical history
13	other data	other data related to the epidemic prevention

**Table 2 healthcare-10-01479-t002:** Key principles and illustrations of the PIPL.

Principle	Illustrations
Lawfulness	Personal information must be processed per the principles of lawfulness, legitimacy, necessity, and good faith, and not in any manner that is misleading, fraudulent, or coercive.
Purpose specification	Personal information must be processed for a clear and reasonable purpose that is directly related to the processing purpose and in a manner that has the most negligible impact on persons’ rights and interests.
Data minimization	Scope limitation: the collection of personal information shall be limited to the minimum scope necessary for the processing purpose and shall not be excessive. Storage limitation: unless any applicable legislation or administrative rule specifies otherwise, the storage duration for personal information shall be the shortest time necessary to achieve the processing objective.
Transparency	Personal information shall be processed under the principles of openness and transparency, with the rules of processing of personal information disclosed and the purposes, methods, and scope of processing expressly stated.
Accuracy	The quality of personal information shall be guaranteed in processing the personal information to avoid adverse impacts on the rights and interests of individuals due to inaccuracy or incompleteness of the personal information.
Accountability	Personal information processors must be accountable for their personal information processing operations
Data security	Personal information processors must take necessary measures (such as encryption and de-identification) to ensure the security of the personal information processed.

## Data Availability

The data presented in this study are available on request from the author.

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
