# Peer review of "Decoding China’s COVID-19 Health Code Apps: The Legal Challenges"

_healthcare, 2022, doi:10.3390/healthcare10081479_

Round 1

Reviewer 1 Report

See PDF attached

Reviewer 2 Report

This is a very well written article that analyses in depth an important issue, to do with a concrete legal analysis of the health code apps operative in China under the extant legal framework that regulates it. The article is very clear on the methods involved and the analysis is convincing and thorough. Whilst the originality of the article is not enormous, this should not be an obstacle for it to be accepted and published. Other than that, there is just one point that could be potentially considered, i.e. issues to do with enforcement of the legal principles discussed (these are not touched upon and incorporating them in the analysis could go a long way towards making a case about how to tackle the legal challenges that the article identifies - who and how could enforce the law whenever there seems to be an issue?). I just also want to add that in line 108 the word should be changed to 'consisting' (instead of 'consisted') and in line 194 'insert' should be changed to 'have'.  

Reviewer 3 Report

First, thank you for the opportunity to review this work. The manuscript is rather diagnostic - the author analyzes the legal challenges of China’s COVID‐19 Health Code Apps. The topic raised in the manuscript is undoubtedly important, especially in the context of public governance, that the author points out to.

However, there are some aspects to improve:

1. The introduction must be substantially improved. The author provides very general information and vague ideas about the topic, in this section, it would be advisable to improve the documentation about this issue, and prior works about the exposed problem. 

3. -       Expand the conclusions a little more, taking up the objectives that were established in the introduction, if any objective was not answered, the reasons that justify why it was not possible to carry out should be given. It is important at this point to show the problems that were faced throughout the investigation.

Considering the previous comments, and although I recognize potential to the ongoing research, I recommend a major revision of the manuscript.

 Other minor suggestions:

1. What are the limitations of research? (for conclusions)

2. What are the possibilities of future research? (for conclusions)

3. - A review of the English wording of the article, as there are some confused phrases.

I enclose several comparative articles dealing with the legal challenges of COVID. In case the author considers taking them into account.

1. Ingrid Sperre Saunes, Karsten Vrangbæk, Haldor Byrkjeflot, Signe Smith Jervelund, Hans Okkels Birk, Liina-Kaisa Tynkkynen, Ilmo Keskimäki, Sigurbjörg Sigurgeirsdóttir, Nils Janlöv, Joakim Ramsberg, Cristina Hernández-Quevedo, Sherry Merkur, Anna Sagan, Marina Karanikolos, Nordic responses to Covid-19: Governance and policy measures in the early phases of the pandemic, Health Policy, Volume 126, Issue 5, 2022.

2. Tracing divergence in crisis governance: responses to the COVID-19 pandemic in France, Germany and Sweden compared Sabine Kuhlmann, Mikael Hellström, Ulf Ramberg, 

Round 2

Reviewer 3 Report

The paper is ready for publication.